# Stable Carbenes as Structural Components of Partially Saturated Sulfur-Containing Heterocycles

**DOI:** 10.3390/molecules27051458

**Published:** 2022-02-22

**Authors:** Alexander B. Rozhenko, Yuliia S. Horbenko, Andrii A. Kyrylchuk, Evgenij V. Zarudnitskii, Sergiy S. Mykhaylychenko, Yuriy G. Shermolovich, Andriy V. Grafov

**Affiliations:** 1Institute of Organic Chemistry, National Academy of Sciences, Murmanska Str. 5, 02094 Kyiv, Ukraine; a_rozhenko@ukr.net (A.B.R.); hannnayul15@gmail.com (Y.S.H.); iamkaant@gmail.com (A.A.K.); ezar@ukr.net (E.V.Z.); misergiy@gmail.com (S.S.M.); sherm@ioch.kiev.ua (Y.G.S.); 2Igor Sikorsky Kyiv Polytechnic Institute, National Technical University of Ukraine, Prosp. Peremohy 37, 03056 Kyiv, Ukraine; 3Department of Chemistry, University of Helsinki, A.I.Virtasen Aukio 1, 00560 Helsinki, Finland

**Keywords:** stable carbenes, ab initio calculations, thiadiazolines, bond elongation

## Abstract

Recently, an unusual elongation of the C-S bond was observed experimentally for some sulfur-containing heterocycles. Using a superior ab initio (SCS-MP2/cc-pVTZ) level of theory, we showed that the phenomenon can be explained by a contribution of a donor–acceptor adduct of a carbene with an unsaturated ligand. One may achieve further elongation of the C-S bond, eventually turning it to a coordinate one, by increasing the stability of each part of the system as, e.g., in the utmost case of spiro adducts with Arduengo carbenes. The effect of carbene stability was quantified by employing the isodesmic reactions of carbene exchange.

## 1. Introduction

Sulfur-containing ring systems comprise an important part of the molecules used in medicinal chemistry. They are listed among the top 100 most commonly used heterocyclic moieties in drug molecules [1,2]. Thiadiazole rings can be found in some antibiotics, such as a sulfonamide sulfamethizole (currently discontinued) and cephalosporin-class antibiotic cefazolin [3], which have entered the World Health Organization’s List of Essential Medicines [4]. Methazolamide [4] and acetazolamide [5] are two carbonic anhydrase inhibitors, containing thiadiazoline and the thiadiazole system, respectively (Figure 1). Moreover, compounds containing the thiadiazole moiety are actively explored as prospective pharmaceutically active ingredients. Additionally, antileishmanial activity was recently demonstrated for quinoline-, thiophene-, and pyrazolo-substituted 1,3,4-thiadiazoles [6,7,8]. Therefore, the research on the preparation, structural characterization, and functional properties of sulfur-containing heterocycles is essential for medicinal chemistry.

Recently, we noticed a significant elongation of the C-S bond [9,10], with respect to the standard value of about 1.83 Å [11], when analyzing X-ray and theoretical data for a series of thiadiazolines in the presence of π-donor R^1^(R^2^) substituent(s) at the carbon atom (Figure 2A). We explained this phenomenon as the result of the high stability of amino-substituted carbenes as structural components of thiadiazoline heterocycles, i.e., the contribution of donor–acceptor adducts (Figure 2B) to the structure of the amino derivatives of thiazolines involving a stable carbene R^1^-C(:)-R^2^. The form B can be considered to be an adduct, where the nitrogen atom forms a covalent bond with carbon and the negatively charged sulfur atom donates to a formally vacant carbon *p*-orbital. In this case, the binding strength depends on the stability of both carbene and thiamine ligand.

In this study, we investigate the C-S bond elongation phenomenon in the broad series of model compounds using quantum chemical ab initio (SCS-MP2/cc-pVTZ) calculations. This theoretical approach is known for a very good level of approximation, which is much better than the classical MP2 method and comparable with much more sophisticated CCSD(T) approach, a “gold standard” for calculations of organic structures [12,13,14,15].

## 2. Results and Discussion

### 2.1. Quantum Chemical Modeling of Sulfur-Containing Heterocycles

Simple symmetrical 2*H*-1,3-dithioles were used as starting compounds. The calculations carried out for structures **1a**–**g** (Figure 3) demonstrated a significant dependence of the C-S bond lengths on the nature of substituents in the position 2 (see Table 1 and Figure 4). For instance, the C-S distance of 1.821 Å in dimethyl-substituted derivative **1a** (Figure 3) is only slightly smaller than the C-S bond length in the tetrahydrothiophene determined in the gas phase (1.839 Å) [11]. This fact is in a good agreement with the low thermodynamic stability of dimethylcarbene Ме-С(:)-Ме. In contrast, the insertion of a dimethylamino group donor lengthens the C-S bond noticeably: the optimized structure **1b** (Figure 3) shows two different C-S bond lengths of 1.852 Å and 1.887 Å. It is well known that highly electronegative and π-electron donor amino groups stabilize the carbenes existing in the singlet ground state [16]. The addition of another dimethylamino group in the position 2 (Figure 3, **1c**) causes further elongation of the C-S bonds to 1.893 Å, while the introduction of an acceptor trifluoromethyl group (**1d**, 1.852 Å and 1.882 Å) does not noticeably affect the lengths of C-S bonds when compared to **1b**. This observation is in agreement with similar stability of the 2-methyl-2-amino- and 2-trifluoromethyl-2-amino-substituted carbenes discussed recently [16].

The short C-S bond in the parent structure **1e** (1.823 Å) conforms to the low stability of triplet methylene. The replacement of the hydrogen atoms by fluorine or chlorine had almost no effect on the C-S bond lengths (1.826 Å and 1.822 Å, respectively).

In contrast, the replacement of sulfur with a more electronegative heteroatom makes the C-S bond significantly longer. In particular, if a nitrogen atom replaces sulfur (Figure 3, compounds **2a**,**b**), the C-S bond lengths reach 1.897 and 1.898 Å, respectively; the insertion of oxygen (**3a**) elongates the bond even further (1.908 Å). Obviously, stronger covalent bonding with nitrogen or oxygen weakens the bond with the remaining sulfur atom. When the methyl group is attached to the endocyclic nitrogen (**2c**), the C-S bond length increases to 1.906 Å (Table 1). On the contrary, the insertion of methyl group at the C=C double bond (**3b**) does not noticeably affect the C-S bond length (1.908 Å in **3a** vs. 1.905 Å in **3b**).

The structures **4a**–**h** (Figure 3) model the experimentally synthesized thiadiazolines [10,11], whereas the structures **5a**–**c** are produced by a substitution of the sp^2^-hybridized carbons by nitrogen atoms. The behavior of the C-S bond within the sub-series **4a**–**d** and **4e**–**h** are similar to the trend discussed above for **1a**–**d**; however, bond lengthening is slightly more pronounced in the series **4** (d(C-S) 1.898 Å and 1.902 Å for **4c** and **4g**, respectively). Obviously, the substitution at the nitrogen molecule does not have any notable effect. Further increase in the number of nitrogen molecules in the heterocycle (Figure 3, **5a**–**c**) leads to a shortening of the C-S bond, which reaches a value of 1.868 Å in **5c**. The replacement of one dimethylamino group with the methyl one (**5b**) further shortens the C-S bond to 1.857 Å, and this bond is even shorter for the dimethyl-substituted derivative **5a** (d(C-S) 1.829 Å).

In order to study the effect of substitution at the C=C double bond in 2*H*-1,3-dithioles, we decided to modify the 2,2-bis(dimethylamino)-substituted structure (**1c**), for which the longest C-S bonds (1.893 Å) were predicted by quantum chemical calculations. Along with the carbene stabilization (Figure 2), the dithione moiety can also be stabilized owing to a specific substitution, leading to the additional elongation of the C-S bond. In the compound **1h** (Figure 5) with two π-donor -NH_2_ groups, the C-S bond length values reach 1.882 Å and 1.891 Å, and thus differ only slightly from the values predicted for the **1c**.

For the sake of comparison, the structure **1i** with two nitro groups has the longest C-S bonds among the compounds under investigation (1.897 Å and 1.919 Å). A difluoro-substituted derivative **1j** also demonstrates elongated C-S bonds with identical lengths (1.903 Å). Evidently, the electronegative fluorine atoms and acceptor nitro groups attached to the C=C double bond increase the lengths of C-S bonds and affect the donor–acceptor character of the whole molecule through the additional stabilization of an isolated dithione molecule. Predictably, a mono-substitution at the C=C double bond (**1k**–**m**) makes the C-S bonds nonequivalent.

Thus, the substitution in sulfur-containing heterocycles significantly affects the C-S bond’s lengths. The range of observed changes is approximately 0.13 Å and substantially depends on the nature and position of the substituents.

### 2.2. Arduengo Carbene-Based Thioheterocycles

Arduengo carbenes are the most thermodynamically stable representatives of the class. Thus, we expected a further elongation of the C-S bond in spiro derivatives **6a**–**g** (Figure 5). A geometrical optimization of the model structures justified our expectations. The values of the C-S bond lengths obtained for series **6** (1.92–2.16 Å) exceed significantly those inherent to other heterocycles discussed in the previous section. Unlike the dithiole **1c**, where both carbon–sulfur bonds were equally elongated, two different types of the C-S bonds were found in the compound **6g** (Figure 5): the covalent bond (1.817 Å) and the coordinate one (2.155 Å). Thus, the **6g** may be best described by the equilibrium structures shown in Figure 5, where the formed imidazolium cation is effectively stabilized by the donor influence of two nitrogen and one sulfur atom, as well as by an additional donation from the coordinated sulfur atom (**6g-B**).

The same applies to the remaining members of the series: the covalent bond is formed with a more electronegative atom (N or O), while sulfur is responsible for the coordinate one. At first sight, the most electronegative atom (oxygen or nitrogen) must carry the negative charge, and not sulfur. However, the formed C-N (or C-O) bond is probably much more favored than the C-S one. A qualitatively similar trend for the C-S bond elongation depending on the nature of the carbene moiety was observed for two series of model linear structures R1(R2)CHSH (**9**) and R1(R2)CHSC(R3)=CH(R4) (**10**) (see ESI, Appendix A). Within the series, poor correlations exist between the calculated C-S distances and total NBO charges located on carbene atoms (see ESI, and Appendix A). Obviously, in line with the growing singlet carbene stabilization, their nucleophilicity and hence, the ability of delocalization of positive charge increase and the C-S bond lengthens.

Interestingly, the C-N bond lengths were almost identical throughout the series **6a**–**g** (1.42–1.45 Å), whereas the corresponding C-S bonds differed significantly from 1.82 Å (**6g**) to 2.27 Å (**6a**). The value obtained for **6c** (2.064 Å) was close to those calculated for **6f** and **6g** (2.034 and 2.155 Å, respectively; see Table 1). Obviously, a conjugation of one of the two lone electron pairs at the chalcogen atom with the formally vacant *p*-orbital of the imidazolium cation would be less dependent on structural distortions and, hence, be more efficient.

### 2.3. Isodesmic Reactions of the Heterocycles with a Carbene Molecule

If the C-S bond length is determined by the stability of the corresponding carbene, then a hypothetical substitution reaction of carbenes in the thioheterocycles **1**–**5** with the more stable carbene **7d** would be either an endothermic (positive ∆H values) or an endergonic process (positive ∆G values). For this purpose, quantum chemical calculations were carried out for the carbenes **7a**–**c** (Figure 6), and their electronic structure is well described within the single-determinant approximation [16].

The ∆H and ∆G values for the first two reactions (Table 2, items 1 and 2) are negative, indicating a stronger binding of the less stable and more reactive trifluoromethyl amino carbene **7b** and difluorocarbene **7c**, which replace more thermodynamically stable Alder carbene **7a [16]**. In contrast, a substitution of the **7a** by the most stable Arduengo carbene **7d** (items 3–9), was less favorable thermodynamically and provided positive values of both ∆H and ∆G. Those results are in a good agreement with the weak C-S covalent bonding predicted for the heterocycles **6a**,**g**.

Interestingly, the proposed theoretical model reproduces well the relative stability in the series of persistent carbenes: the Alder carbene vs. saturated and unsaturated Arduengo carbenes. The endothermic (endergonic) effects for the corresponding reactions (Table 2, items 10–14) were only approximately one-half of those found for the reactions with unsaturated Arduengo carbene (Table 2, items 3–9). Therefore, the relative stability of carbenes increases in the series **7d** > **7e** > **7a**. Structures **8a**–**e** (Figure 6) based on the saturated carbene **7e** demonstrate intermediate C-S bond lengthening comparable to those found for the species involving the less stable Alder carbene (~1.89 Å, Table 1).

## 3. Materials and Methods

All calculations were carried out using the TURBOMOLE program package (version 6.4 and 7.5) [17,18]. Geometrical optimization and the calculation of the ΔE, ΔH, and ΔG values were performed using SCS-MP2 level of approximation [12,19,20,21] with triple-ζ cc-pVTZ Dunning’s basis sets [22]. Resolution of the Identity (RI) approximation [23,24] was utilized in all cases to increase calculation speed and efficiency. The main structure parameters are given in Table 1. All energy values and Cartesian coordinates for the optimized structures are presented in the Appendix A.

Vibration frequencies and corrections for calculation of relative energies and relative Gibbs free energies were derived numerically at the SCS-MP2/cc-pVTZ level of theory. All the optimized structures corresponded to local energy minima, and no imaginary frequencies were detected by the vibration analysis. In order to derive ΔE magnitudes (Appendix A), the corresponding corrections on vibrations at 0 K (ZPE) were added to the total energy values. For relative enthalpy change (ΔH) and Gibbs free energy values (ΔG), the corresponding corrections for total energy values were calculated under standard conditions (pressure 0.1 Pa, temperature 298.15 K) and scaled at 0.95.

NBO charge calculations were carried out at the SCS-MP2/cc-pVTZ level of ap-proximation using the optimized geometries and the NBO procedure [25,26,27] imple-mented into the TURBOMOLE program.


The Jmol [28,29] program was used for the graphical presentation of the structures.

## 4. Conclusions

A series of sulfur-containing heterocyclic compounds were studied using quantum chemical calculations at the ab initio (RI-SCS-MP2/cc-pVTZ) level of approximation. A noticeable elongation of the C-S bond was observed in the cases of stable carbenes (Alder or Arduengo carbenes). The phenomenon is also affected by the nature of other heteroatoms in the molecules and the substitution character. Both experimentally observed and theoretically predicted elongations of the C-S bonds in the heterocycles under investigation were in a good agreement with the representation of the molecules as donor–acceptor complexes of carbenes with ligands (Figure 2B). On one hand, the longest C-S bonds were found for Arduengo carbene derivatives, and in the frontier cases, the covalent bond transformed into a weak coordinate bonding. On the other hand, the presence of substituents stabilizing the isolated dithione moiety (a second counterpart of the imaginary donor–acceptor complex) contributed to further elongation of the C-S bonds.

## Figures and Tables

**Figure 1 molecules-27-01458-f001:**
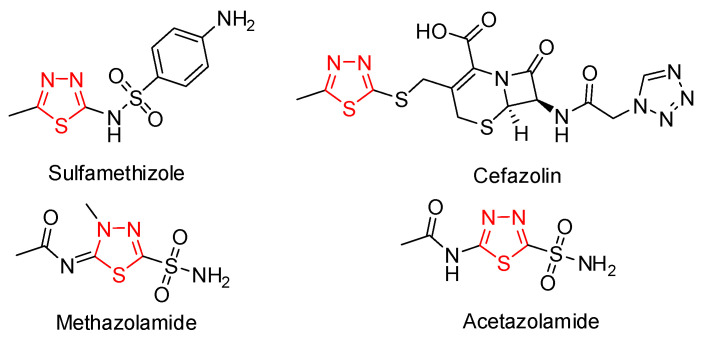
Thiadiazole-containing drugs.

**Figure 2 molecules-27-01458-f002:**
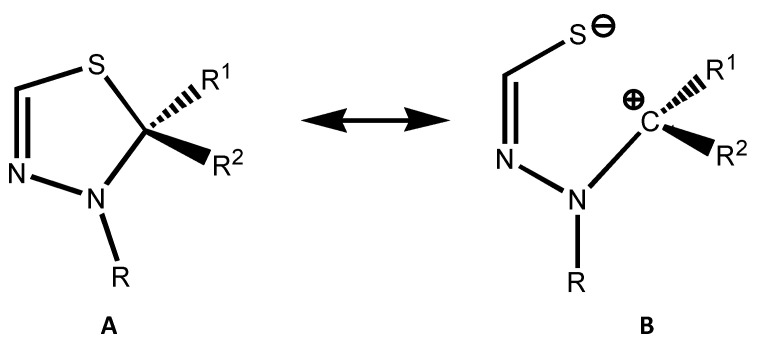
Two canonical forms in thiadiazolines involving covalent (**A**) and coordinate (**B**) C-S bonds.

**Figure 3 molecules-27-01458-f003:**
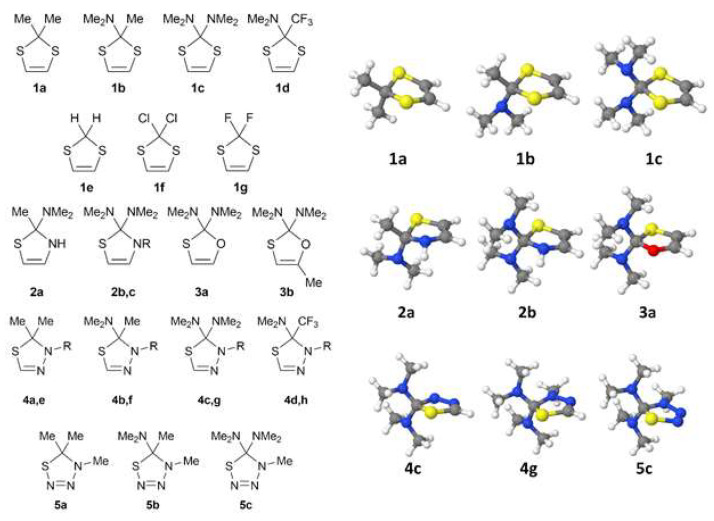
Model structures **1a**–**g**, **2a**–**c**, **3a**,**b**, **4a**–**h**, and **5a**–**c**; R: H (**2b**, **4a**–**d**) and Me (**2c**, **4e**–**h**) (**left**). Jmol presentation of equilibrium structures (**right**). Color coding: C: gray, H: white, N: blue, S: yellow.

**Figure 4 molecules-27-01458-f004:**
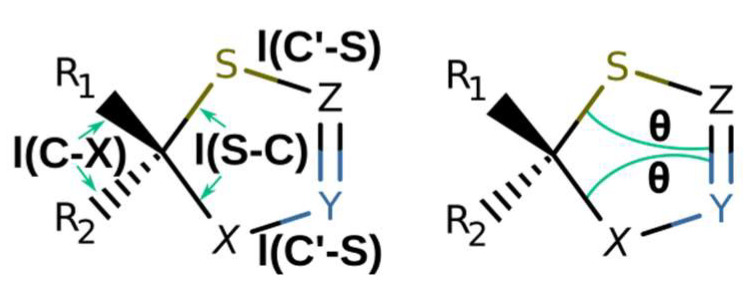
Main structural parameters of optimized structures **1**–**6** (Table 1).

**Figure 5 molecules-27-01458-f005:**
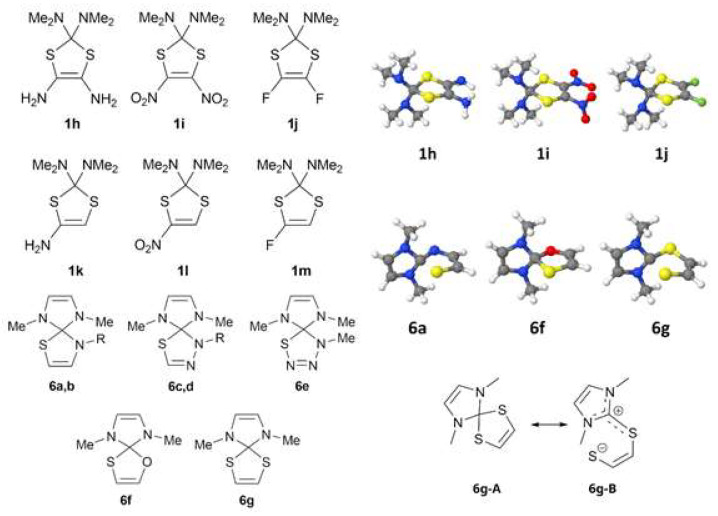
Model structures **1h**–**m** and **6a**–**g** (**6a**,**c**: R=H, **6b**,**d**: R=Me) (**left**). Jmol presentation of equilibrium structures (**right**). Color coding: C: gray, H: white, N: blue, O: red; F: green; S: yellow. Two polar forms of the compound **6g** (**bottom right**).

**Figure 6 molecules-27-01458-f006:**
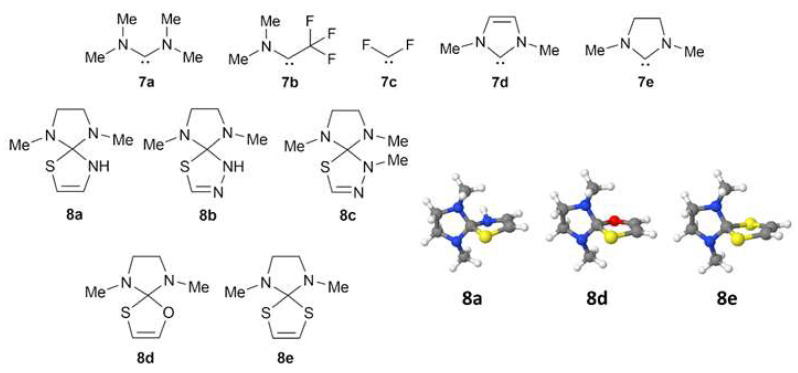
Model structures **7a**–**e** and **8a**–**e** (**left**) and the optimized (RI-SCS-MP2/cc-pVTZ) structures **8a**,**d**,**e** (**right**). Color coding: C: gray, H: white, N: blue, O: red; S: yellow.

**Table 1 molecules-27-01458-t001:** Main structure parameters of optimized structures **1**–**6** (see Figure 4 for designations).

Structure	l(S-C), Å	l(C′-S), Å	l(C-X), Å	*θ*, Degrees
**1a**	1.821	1.779	1.544	19.4, −19.4
**1b**	1.852, 1.887	1.752	1.444 (C-N), 1.532 (C-C)	15.8, −13.5
**1c**	1.893	1.747	1.446	8.0, −5.0
**1d**	1.852, 1.882	1.750, 1.751	1.439 (C-N), 1.552 (C-C)	12.2, −8.9
**1e**	1.823	1.763	1.087, 1.088	19.3, −19.3
**1f**	1.822	1.755	1.779, 1.808	19.4, −19.4
**1g**	1.826	1.750	1.353, 1.357	7.9, −7.9
**1h**	1.882, 1.891	1.757, 1.773	1.445, 1.447	15.0, −12.4
**1i**	1.897, 1.919	1.728, 1.744	1.426, 1.435	14.8, −12.6
**1j**	1.903, 1.903	1.752, 1.752	1.436, 1.438	13.8, −11.7
**1k**	1.886, 1.890	1.755, 1.761	1.444, 1.447	13.2, −13.2
**1l**	1.898, 1.907	1.727, 1.744	1.437, 1.439	9.5, -8.1
**1m**	1.897, 1.899	1.750, 1.751	1.439, 1.442	11.6, −8.8
**2a**	1.897, 1.467 (C-N)	1.765, 1.400 (C-N)	1.452 (C-N), 1.527 (C-C)	15.9, −9.9
**2b**	1.898, 1.467 (C-N)	1.761, 1.398 (C-N)	1.450, 1.459	11.3, −5.4
**2c**	1.906, 1.462 (C-N)	1.760, 1.383 (C-N)	1.447, 1.457	6.0, −12.6
**3a**	1.908, 1.446 (C-O)	1.755, 1.366 (C-O)	1.434, 1.437	0.5, −2.8
**3b**	1.905, 1.442 (C-O)	1.756, 1.372 (C-O)	1.435, 1.439	0.4, −2.5
**4a**	1.846, 1.479 (C-N)	1.764, 1.401 (N-N)	1.522, 1.529	15.6, −28.5
**4b**	1.888, 1.474 (C-N)	1.756, 1.391 (N-N)	1.444 (C-N), 1.527 (C-C)	22.8, −11.6
**4c**	1.898, 1.475 (C-N)	1.751, 1.384 (N-N)	1.438, 1.458	7.5, −20.0
**4d**	1.853, 1.470 (C-N)	1.756, 1.391 (N-N)	1.446 (C-N), 1.551 (C-C)	−0.5, −8.0
**4e**	1.847, 1.482 (C-N)	1.760, 1.396 (N-N)	1.523, 1.532	27.7, −16.1
**4f**	1.888, 1.471 (C-N)	1.755, 1.382 (N-N)	1.444 (C-N), 1.529 (C-C)	22.9, −12.7
**4g**	1.902, 1.474 (C-N)	1.749, 1.370 (N-N)	1.441, 1.451	9.4, −20.0
**4h**	1.858, 1.465 (C-N)	1.754, 1.380 (N-N)	1.459 (C-N), 1.559 (C-C)	0.8, −5.8
**5a**	1.829, 1.476 (C-N)	1.761(S-N), 1.368 (N-N)	1.525, 1.531	14.8, −24.3
**5b**	1.857, 1.466 (C-N)	1.752(S-N), 1.358 (N-N)	1.450 (C-N), 1.530 (C-C)	20.1, −12.9
**5c**	1.868, 1.471 (C-N)	1.739(S-N), 1.346 (N-N)	1.447, 1.451	8.0, −15.7
**6a**	2.272, 1.425 (C-N)	1.763, 1.414 (C-N)	1.409, 1.413	5.0, −13.2
**6b**	1.990, 1.440 (C-N)	1.758, 1.396 (C-N)	1.429, 1.436	9.0, −16.7
**6c**	2.064, 1.445 (C-N)	1.740(S-N), 1.395 (N-N)	1.408, 1.412	25.6, −11.3
**6d**	2.045, 1.445 (C-N)	1.740(S-N), 1.389 (N-N)	1.412, 1.418	25.2, −11.6
**6e**	1.919, 1.448 (C-N)	1.739(S-N), 1.356 (N-N)	1.437, 1.439	8.4, −16.8
**6f**	2.034, 1.409 (C-O)	1.750, 1.376 (C-O)	1.411, 1.411	0.0, 0.0
**6g**	2.155, 1.817	1.737, 1.755	1.399, 1.401	12.4, −14.7
**8a**	1.890, 1.467 (C-N)	1.760(C-S), 1.401 (C-N)	1.448, 1.450	14.1, −9.2
**8b**	1.886, 1.457 (C-O)	1.756(C-S), 1.353 (C-O)	1.425, 1.431	4.6, −6.0
**8c**	1.885, 1.891	1.747, 1.747	1.432, 1.438	7.6, −12.7
**8d**	1.894, 1.472 (C-N)	1.748(C-S), 1.388 (N-N)	1.448, 1.450	18.8, −9.5
**8e**	1.895, 1.472 (C-N)	1.747(C-S), 1.380 (N-N)	1.436, 1.443	20.6, −11.3

**Table 2 molecules-27-01458-t002:** Enthalpy change (ΔH) and Gibbs free energy (ΔG) values in kcal/mol for isodesmic reactions (items 1–14).

Item	Reaction	ΔH	ΔG
1	**1c** + **7b** → **1d** + **7a**	–5.2	(–3.3)
2	**1c** + **7c** → **1g** + **7a**	–8.8	–10.0
3	**2b** + **7d** → **6a** + **7a**	23.2	22.2
4	**2c** + **7d** → **6b** + **7a**	20.6	19.8
5	**4c** + **7d** → **6c** + **7a**	21.8	20.7
6	**4g** + **7d** → **6d** + **7a**	19.6	18.6
7	**5c** + **7d** → **6e** + **7a**	22.3	21.3
8	**3a** + **7d** → **6f** + **7a**	26.8	25.1
9	**1c** + **7d** → **6g** + **7a**	24.2	22.6
10	**2b** + **7e** → **8a** + **7a**	8.9	9.1
11	**4c** + **7e** → **8b** + **7a**	8.4	8.5
12	**4g** + **7e** → **8c** + **7a**	7.1	7.2
13	**3a** + **7e** → **8d** + **7a**	11.1	11.2
14	**1c** + **7e** → **8e** + **7a**	9.8	10.1

## Data Availability

The data supporting the conclusions of this article are included within the article and Supplementary materials.

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
