# Peer review of "Stable Carbenes as Structural Components of Partially Saturated Sulfur-Containing Heterocycles"

_molecules, 2022, doi:10.3390/molecules27051458_

Round 1

Reviewer 1 Report

See attached file

Author Response

Dear Reviewer,

We thank you for your effective evaluation and interesting suggestions.

Generally, the concept of “polar” (or “mesomeric”) structures has been used for many years. For a long time, it served for a description of changes in structure and reactivity of organic compounds. The concept is very conditional and cannot be used instead of strict quantum chemistry calculations, but only for a more clear illustration of the calculation results. The Reviewer considers the structure B as a simple combination of the starting materials reacting via a transition state. However, the figure shows the situation when the reagents already passed the transition state and the atom positions in B are close to those in A.

Nevertheless, we admit that the structure B does not exactly reflect the C-S bond elongation and modified the Figure (see Figure 2).

1) Last sentence before Figure 2. The described formal double donation from LPs at S and N to a vacant P atomic orbital (AO) is impossible, because this would locate the filled carbene sp2 AO pointing to the ring centre, which is rather unlikely to occur.

The Reviewer’s remark is reasonable. The text (marked in yellow) was revised in accordance with the modified Figure 2.

2) Paragraph after Figure 2. SCS-MP2/cc-pVTZ is a good computational level for geometry optimizations and reasonable for energy evaluations, but it is not superior to CCSD(T) with an appropriate basis set.

The authors did not affirm that SCS-MP2/cc-pVTZ is superior to CCSD(T). In order to avoid ambiguity we have slightly changed the text (lines 50-52).

3) Page 2, line 69-70. “2-methyl-2-amino-”. And remove “in” after “recently”.

Corrected.

4) p2, L71-73. Why replacement with two F or Cl atoms has almost no effect on the C-S bond length. Please suggest an explanation. Why the authors did not study the related dibromoderivative?

The substitution effect was not observed. Generally, this is in agreement with quite low stability of difluoro- and dichlorocarbenes. As the matter of fact, a presence of an effect is always better to explain than the absence of it. Therefore, there is no reason to further expand the series, investigating the dibromoderivative.

5) p3, L81-82. If electronegative groups elongate the C-S bons (as stated at the end of page 2), why N-methylation (N becoming less electronegative due to the +I effect of the methyl group) has the same effect? The same inconsistency appears in L89 when comparing 4c and 4g. And why Cl2- and F2-disubstitution (1f-g) has almost no effect? This seems contradictory.

The easiest explanation for the small difference in the C–S bond lengths for N-Me derivatives vs. NH-heterocycles consists in an overall little effect of the methyl group on the electronegativity of nitrogen. Thus, the C-S bond elongation was not found.

As far as Cl2- and F2-disubstitutions are concerned, please see above.

6) p5, L112-113. If molecules 1i,j have donor-acceptor (diad) character, some intramolecular electron transfer (IET) should be expected and this would originate a NIR optical transition band. Have this ever been experimentally measured or computationally (TD-DFT) proven? At least the latter should be checked.

The compounds were studied theoretically, and not synthesized, their spectral properties are beyond the scope of the manuscript.

7) Figure 3 and 5. I would suggest the authors to initially explore the effect of the R1 and R2 “carbene” substituents on even more simple model acyclic compounds having the R1R2C unit connected to an SH (I) or S-vinyl (II) group (see figure). In my hands, at a slightly lower computational level and using nine I-type and seven II-type derivatives, a rather acceptable linear correlation (R2 = 0.763) is obtained between the C-S bond distance and the Mulliken electric charge at the “α” (R1R2CH) unit. This would support a contribution of hyperconjugation structures similar to that depicted as 6g-B in Figure 5. I encourage the authors to check this fundamental computational level. And preferably using natural charges.

The authors understand the idea of the Reviewer to use a comparable series of compounds to find out, whether the observed effect has more general character or not. However, the model structures proposed are very different from those studied in the present work, they would not provide new data supporting the explanation proposed. In addition, the regression coefficient of 0.763 is too small to speak about an existing trend.

8) p5, L123-124. In addition to electronic effects, some other factors can sum up to explain this extra C-S bond elongation. Consider for instance the extra RSE (ring strain energy) introduced in the 5-membered S-containing heterocycle (6) by the spiro-connected “NHC ring” (see the Streubel’s group report on Dalton Trans., 2013, 42, 8897).

The authors hypothesized on the nature of the C-S bond elongation and tried (upon a suggestion of Prof. Didier Bourissou) to investigate compounds from the series 6 to make the effect more pronounced. However, it seems that the S-containing heterocycles (series 6) with the spiro-connected “NHC ring” demonstrate either very long C-S bond or even dissociation of the bonds. That fact supports the initial version on the dependence of the C-S bond length on the carbene stability.

Concerning the influence of the spiro-structure on the ring strain and the paper suggested for citation, the authors presume that a comparison of the structures 6 under investigation, which have no ring strain, with bicyclic systems involving a highly strained three-membered would not be correct. Nevertheless, the authors are ready to cite the proposed references, if the Reviewer insists.

9) p5, L126-127. The two different (covalent and dative) C-S bonds should be properly characterized according to appropriate differentiating bond descriptors (e.g. Inorg. Chem, 2020, 59, 12829).

Again, in the authors’ opinion, there is a lot of ways to characterize chemical bonds. However, the quantitative characterization of the C-S bond strength is also beyond the scope of the manuscript. That kind of data would only increase unreasonably the amount of work, while the chemical essence would be drown in an array of numbers.

10) p5, L131-133. This would unfavour hyperconjugative (charge separated) structures like 6g-B due to the natural tendency of the most electronegative atom to stabilize negative charges and of the least electronegative atom to donate its LP to stabilize the positive charge. Some explanation is required.

The Reviewer’s remark is reasonable. The requested explanation is provided on the p. 6 (lines 134-136).

11) p6, L135. “The C-S distance obtained for 6c (2.064 Å) and 6d (2.045 Å) were close to those computed (or obtained, not predicted) for 6f ...

Correction accepted.

12) p6, L136. “... conjugation of one of the two lone ...”

Correction accepted.

13) p6, L138-139. More efficient than the conjugation of the LP at N in 6c,d-B? Why?

Corrected, see p. 6 (lines 140-142).

14) p6, L143. Compound 6 is not a carbene.

Corrected to 7d.

15) p7, L160-163. This is expected by just assuming the lower stability of 7b,c compared to 7a.

And it is clear that the assumption as well as the authors’ hypothesis are supported very well by the calculations.

16) p7, L163-166. Same for 7a compared to 7d.

Please, see the response 15.

17) Table 2 and 3 should be merged and equations 1-14 placed as entry in (the new) Table 2.

Correction accepted.

Reviewer 2 Report

The authors have presented theoretical studies about C-S bond elongation in the series of sulfur-containing heterocyclic compounds. Its elongation has been associated with the formation of stable carbenes. This is an interesting paper.

Please find in the attached file the corrections that are necessary before publication. 

Author Response

Dear Reviewer,

thank you very much for your work and suggestions.

We gratefully accepted all corrections you proposed.

Round 2

Reviewer 1 Report

See attached file

Author Response

Dear Editor,

We thank the Reviewer 1 for the additional suggestions, which are partially taken into account in the revised version of the manuscript.

I disagree with the use of “polar” within this context, in line with the IUPAC definition, and should be named “contributing” (or “resonance”) structures or “canonical forms” insteadTherefore, only structure B is polar in Figure 2, and the caption must be changed accordingly

Accepted and the caption was corrected.

Next, I proposed the authors to study the electronic effect of the C-substituents (R1 and R2), parametrised by the electric charge at C, in the C-S bond elongation in acyclic model compounds. Maybe the authors misunderstood the proposal, but this (or similar “experiment”) is the only way of studying the origin of the observed C-S bond elongation. There is no need of evoking a carbene stability for explaining the C-S bond-elongation, because due to the Hammond’s postulated they should not be directly connected. Instead, the direct effect between electric charge at C (i.e. electronic effects of C-substituents) on C-S bond elongation (or similar DIRECT connection) must be analysed. And maybe this could also be related to a similar direct relationship in the corresponding carbenes R1R2C: (or maybe not). In other words, the proposed indirect connection between stability of carbenes (inferred by means of isodesmic reactions) and C-S elongation in the “adducts” is (strictly speaking) not acceptable, unless the individual effect(s) affecting both (for instance electronic effects of substituents) can be related.

The authors performed additional calculations for the model structures 9 and 10, required by the Reviewer. The corresponding data are collected in the ESI (see Figure S1 and Table S2). The trend found is in a good agreement with the conclusions made for the series of sulfur-containing heterocycles.

Finally, characterizing the C-S bond as covalent or dative is, by no means, meaningless. Indeed, the core of this manuscript is dealing with differently elongated C-S bonds ranging from purely covalent to essentially dative, as deduced from the collected bond distances. Therefore, classification of these bonds in one or another category (or borderline cases) should be mandatory, and this requires parametrization, at least for some most representative cases.

The authors do not agree with the Reviewer’s opinion. The bond distances strongly exceeding the typical value are usually considered as dative or coordinative. This approach is clearly understood by chemists. Again, any quantitative classification of the bonds would only unnecessarily overload the content of the manuscript.